# Metal-Based Nanostructured Therapeutic Strategies for Glioblastoma Treatment—An Update

**DOI:** 10.3390/biomedicines10071598

**Published:** 2022-07-05

**Authors:** Agata M. Gawel, Ravi Singh, Waldemar Debinski

**Affiliations:** 1Histology and Embryology Students’ Science Association, Department of Histology and Embryology, Faculty of Medicine, Medical University of Warsaw, Chalubinskiego 5, 02-004 Warsaw, Poland; agata.gawel@student.wum.edu.pl; 2Department of Cancer Biology, Wake Forest School of Medicine, Winston-Salem, NC 27157, USA; rasingh@wakehealth.edu; 3Brain Tumor Center of Excellence, Wake Forest Baptist Medical Center Comprehensive Cancer Center, Winston-Salem, NC 27157, USA

**Keywords:** glioblastoma, metal-based nanocarriers, combinatory therapy, theranostic nanocarriers

## Abstract

Glioblastoma (GBM) is the most commonly diagnosed and most lethal primary malignant brain tumor in adults. Standard treatments are ineffective, and despite promising results obtained in early phases of experimental clinical trials, the prognosis of GBM remains unfavorable. Therefore, there is need for exploration and development of innovative methods that aim to establish new therapies or increase the effectiveness of existing therapies. One of the most exciting new strategies enabling combinatory treatment is the usage of nanocarriers loaded with chemotherapeutics and/or other anticancer compounds. Nanocarriers exhibit unique properties in antitumor therapy, as they allow highly efficient drug transport into cells and sustained intracellular accumulation of the delivered cargo. They can be infused into and are retained by GBM tumors, and potentially can bypass the blood–brain barrier. One of the most promising and extensively studied groups of nanostructured therapeutics are metal-based nanoparticles. These theranostic nanocarriers demonstrate relatively low toxicity, thus they might be applied for both diagnosis and therapy. In this article, we provide an update on metal-based nanostructured constructs in the treatment of GBM. We focus on the interaction of metal nanoparticles with various forms of electromagnetic radiation for use in photothermal, photodynamic, magnetic hyperthermia and ionizing radiation sensitization applications.

## 1. Introduction

According to the GLOBOCAN World Health Organization (WHO) statistics, cancer incidence is on the rise with 19.3 M new cancer cases and nearly 10.0 M cancer deaths reported worldwide in 2020. Tumors of the brain and nervous system (CNS) affected 308,102 people around the globe in 2020 [1]. The most frequently diagnosed primary, malignant brain tumor is glioblastoma (GBM), a rapidly growing cancer that aggressively invades surrounding normal brain tissue [2].

GBM is a subset of gliomas that are classified by four grades based on the rate of spread, cell shape and size, genetic markers and the presence of necrosis. Grades I and II are classified as low-grade gliomas, while grades III and IV (high-grade glioma) are categorized as GBM [3]. The incidence rate of GBM worldwide is estimated to be from 0.6 to 5 per 100,000 while in the United States about 3 people per 100,000 are diagnosed with this tumor type [4,5]. The 5-year survival rate for GBM patients is <5% [6]. GBM is more prevalent in males with a 1.6-fold higher incidence rate than in females [7]. GBM is typically observed in two age groups of patients: older than 65 years, who are most often diagnosed with de novo GBM without prior evidence of a lower grade tumor (primary GBM), and younger patients of an average age of 45 years, who may be diagnosed with GBM after the progression of lower grade tumors (secondary GBM) [8]. Although they are difficult to distinguish based upon histologic evidence, primary and secondary GBMs differ in underlying genetics and oncogenic drivers [9]. 

The standard therapy consists of surgical removal of the tumor, which is usually supported by chemotherapy and radiation [10]. The major routes of drug delivery include oral or intravenous infusion; thus, side effects are common, as systemic drugs are toxic to peripheral organs [11]. Currently available treatments prolong patients’ lives up to 15–17 months and the disease is almost always fatal. Although GBM constitutes only 2–3% of all cancer cases, more patients die from malignant gliomas than from melanoma [12].

The molecular players and mechanisms governing the transformation and spread of GBM tumors are complex and remain to be fully elucidated. Mutations leading to the the downregulation of tumor suppressors (TP53, PTEN) and overexpression of oncogenes (EGFR, PIK3CA) contribute to the formation and advancement of GBMs. Moreover, it is suggested that interplay between signaling cascades (e.g., AKT, MAPK, NF-κB), focal adhesion proteins (including FAK kinase, paxillin), extracellular matrix proteins (e.g., integrins, metalloproteinases) and various environmental factors promotes the development and progression of GBM cells [5]. 

Despite significant efforts to develop new therapies, one of the key barriers in the treatment of cancers is inherent or acquired drug resistance, a characteristic of many GBMs, which significantly limits the efficacy of drug-based treatments [13]. In GBMs treated with alkylating agents such as temozolomide (TMZ, the standard of care drug used in the treatment of GBM) [14], resistance frequently results from alkylation of guanine at the O^6^ position caused by the overexpression of O^6^-methylguanine methyltransferase (MGMT) [15]. Other mechanisms of drug resistance include hyperactivation of transport proteins belonging to the ATP-binding cassette transporter family, including the ABCB1 protein (P-glycoprotein; P-gp), which plays a key role in the removal of drugs from cells in many tumors [16,17]. 

Another significant obstacle in GBM treatment is the blood–brain barrier (BBB) [18]. This natural physico-biological protection of the CNS significantly restricts the access of drugs to the CNS, which limits the translational potential of most experimental therapies shown to have substantial activity on GBM tumor cells in vitro [19,20]. The efficacy of treatment is further restricted due to the heterogeneity, immunosuppressive abilities and infiltrative nature of GBM cells [21]. There is a pressing need for innovative treatment strategies for GBM patients that can overcome drug resistance, increase the delivery of therapeutics to the tumors and reduce off-target toxicity. A promising approach is the use of nanostructured therapeutics of various classes (Figure 1). 

Nanosized vehicles can efficiently deliver chemotherapeutics, which enables sustained accumulation of drugs and the killing of tumor cells [22]. When equipped with designated ligand(s), they are capable of specifically targeting tumor cells, potentially reducing damage to normal (non-cancerous) cells [23]. The advantages of nanoparticles in general stem from their high cargo loading capacity (binding of both hydrophilic and hydrophobic drugs), controlled (prolonged) drug release capability and potential to cross the BBB [24]. In addition to these abilities, the unique interactions of metal nanoparticles with various forms of electromagnetic radiation enables their use in photothermal, photodynamic, magnetic hyperthermia and ionizing radiation sensitization applications. Classic nanoparticle-based drug delivery vehicles based upon phospholipid vesicles (e.g., liposomes), polymers or hydrogels for GBM treatment have been extensively reviewed elsewhere (e.g., [25,26]). Herein, we review recent progress (2018–2021) made in the field of metal-based nanotherapeutic strategies.

## 2. Metal-Based Nanoparticles

Metal-based nanoparticles (NPs) are among the most promising tools in the therapy of various cancers. The focus of recent studies in the field of metal-based NPs has been on: (i) gold nanoparticles; (ii) silver nanoparticles; (iii) iron oxide nanoparticles; (iv) superparamagnetic iron oxide nanoparticles; (v) non-iron-based magnetic nanoparticles; and (vi) quantum dots (Figure 2 and Table 1). They present the capability to traverse the BBB and can effectively distribute both drugs and other therapeutic compounds, including nucleic acids (e.g., microRNA), at the tumor site. Metal-based NPs have been reported to not only reduce GBM progression, but also lower radioresistance of cancer cells [27].

Generally, metal NPs may be physically synthesized either using the “bottom-up” (constructive method, assembling from atomic level with the usage of molecules or clusters) or “top-down” (destructive method, by breaking up bulk material) approaches. The “bottom-up” methods are considered to be more precise and affordable. Electrochemical or radiation-induced methods are used for the synthesis of metal NPs. One of the most novel strategies is the formulation of metal NPs based on microemulsions. The reduction in metal ions from their ionic salts is achieved by the usage of reducing agents in the presence of a stabilizing agent under favorable, but likely moderate, environmental conditions, without using elevated temperature or pressure. Importantly, the generated pools of NPs are relatively highly homogeneous in composition, size and shape under the optimized synthesis conditions. Among reducing agents, sodium borohydrate, sodium citrate, tannic acid, hydrazine, hydrogen and alcohols are most commonly used. This method also offers the usage of the so-called “green materials”, which also require moderate preparation conditions. Green materials can act as reducing and/or stabilizing agents. They are recognized as non-toxic and environmentally friendly, and include bacteria, fungi and enzymes. Therefore, the synthetized NPs can be more effective as their toxicity is restricted [28,29].

In contrast, the synthetic procedure in the “top-down” methods requires more harsh conditions, including high pressure and temperature making this approach less affordable. Moreover, the structure of the surface of delivered NPs is irregular, which limits physical and chemical properties of the delivered NPs. The most commonly used “top-down” methods are sputtering, micropatterning, milling or laser ablations. The efficiency of these methods depends on various factors and special equipment for processing is additionally required [29]. 

### 2.1. Gold Nanoparticles (AuNPs) 

Gold nanoparticles (AuNPs) have received a great deal of attention in cancer treatment over the last years. Formation of these NPs first requires a reduction of a gold precursor to achieve AuNPs, which are then stabilized using a capping agent (preventing their agglomeration) [30]. AuNPs may be physically synthesized either using the “bottom-up” or “top-down” method. Additionally, they may be produced using the chemical approach or biological materials can be applied as a source of AuNPs (i.e., bacteria, fungi, plants or algae) [30]. The main route of cellular uptake of AuNPs is believed to be through endocytosis [31,32]. The size of AuNPs determines their uptake by tumors and permeation of the BBB [33]. It has been reported that the use of smaller (10 nm) AuNPs is more suitable (Kang et al., 2019), whereas other studies have shown that AuNP-based nanocarriers with a size of 70 nm might display the best properties for crossing the blood–brain tumor barrier (BBTB) [34]. Alongside promising drug delivery properties, Au-based nanocomposites present satisfactory biocompatibility, as evidenced by a transient increase in drug levels in tumors relative to the time of exposure to therapy [35]. Interestingly, certain particles that can be used for both chemo- and photothermal therapy, such as the loquat-shaped Janus carrier (a particle with an anisotropic structure), also present the potential to spontaneously exit the tumor site, which may prevent long-term side effects of treatment [35]. 

In order to enhance the functionality and increase the drug delivery capability of AuNPs, various modifications or “decorations” of their structure, shape or surface have been explored (Table 1). AuNPs with an irregular surface (gold nanourchins) have been proposed as delivery vehicles for cytotoxic agents, including celastrol [36]. These nanocarriers may allow modifications of the structure of GBM cells by increasing the level of ROCK1 (a ROCK family kinase), responsible for cytoskeleton remodeling [36,37]. Albertini et al. (2019) investigated peptide (i.e., arginine-glycine-aspartic acid)-decorated AuNPs, which were administered to mice. Such altered AuNPs presented 3.7-fold and 1.2-fold higher accumulation in the intracranial tumor model, in comparison to free-AuNPs or polyethylene glycol (PEG)ylated AuNPs, respectively [38]. 

AuNPs may also be used as labels for magnetic resonance imaging (MRI) and photoacoustic visualization of the tumor, for targeted single photon emission computed tomography (CT) imaging or as prognostic tools [39,40,41,42]. Theranostic application of labeled polyethylenimine-entrapped AuNPs was recently reported for radionuclide therapy [43,44]. As described below, coating of AuNPs is also an effective method for improving radiosensitivity and BBB permeability.

#### 2.1.1. Application of AuNPs for Overcoming the Blood–Brain Barrier

The BBB and BBTB are two critical obstacles in the successful passage of nanocarriers into GBM cells. The BBTB, which is an abnormal, fenestrated and heterogenous variant of the BBB, permits the passage of contrast agents (allowing magnetic resonance imaging of GBM), although the permeability of drugs varies. It is assumed that the BBTB may support drug influx at the site of the tumor; nevertheless, it has also been shown that the accumulation of chemotherapeutics is often irregular, therefore limiting the efficacy of treatment [45,46]. Some of the proposed methods of overcoming this barrier include the usage of focused ultrasound techniques and microbubbles for mechanical opening of the BBB. Another strategy is the disruption of the BBB via alteration of the expressional profile of genes encoding for proteins controlling tight junction complexes [47]. 

Several recent studies have considered in situ assembly of nanoplatforms and arming them with specific ligands that permit passage through these barriers and entering the tumor site. An example of such a nanoplatform is the cluster built of RGD peptide-modified bisulfite-zinc^II^-dipicolylamine-Arg-Gly-Asp (Bis(DPA-Zn)-RGD) and ultra-small Au-indocyanine green (ICG) NPs. Such a construct can be precisely delivered to the tumor due to its positive charge and neovascular targeting abilities exhibited by ICG (a cyanine dye commonly applied for the imaging of retinal blood vessels [48]). ICG, when exposed to light, can act as a generator of reactive oxygen species (ROS) and damage cancer cells, effectively inhibiting tumor progression by up to 93.9% [49].

An alternative to in situ self-assembling nanocarriers for overcoming the BBB is the intranasal administration of metal-based NPs. The nose-to-brain (N2B) strategy is recognized as one of the most effective approaches [50]. Nevertheless, physiological barriers for N2B delivery also exist, including efflux transporters, nasal metabolism, mucociliary clearance, surface area of the olfactory region and presence of drug-specific target receptors/transporters [51,52,53]. Yin et al. (2020) reported a N2B administered nanoformulation that effectively targeted GL261 GBM cells in a murine model. Methoxypolyethylene glycol-detachable (d)-polyethyleneimine (mPEG-d-PEI)-coated AuNPs administered together with TMZ bypassed the BBB and induced an immune response in situ leading to immunogenic cancer cell death [54]. Another concept for the improvement of the intranasal route, demonstrated by Wang and collaborators (2021), is the application of antibody-conjugated NPs. They developed and tested TMZ-AuNPs conjugated to an antibody against the ephrin type-A receptor 3 (anti-EphA3-TMZ@AuNPs). It was found that this bioconjugate successfully passes the BBB to reach the cancerous cells in the brain tissue of orthotopic glioma-bearing rats. Additionally, the glioma C6 cells treated with anti-EphA3-TMZ@AuNPs presented greater uptake of the drug, and thus its toxic effect on cells increased. At the same time, the intranasally delivered formulation did not accumulate in the normal tissues of the treated animals (murine model) and lower drug resistance was also observed, as the nanocomplexes downregulated the expression of MGMT. Most importantly, the tested rats treated with anti-EphA3-TMZ@AuNPs were found to have a >2-fold prolonged survival time, in comparison to the control groups administered saline or TMZ alone [55]. 

A different approach to overcoming the BBB was described by Pall et al. (2019), who proposed the implementation of AuNP-loaded macrophages for effective bypassing of the BBB and subsequent targeting of GBM cells. After loading the nanoparticles into macrophages, enhanced delivery (by 2-fold) of AuNPs was achieved. They observed a positive correlation between the higher expression of NHE9 (an endosomal Na^+^/H^+^ exchanger typically overexpressed on GBM cells) and increased internalization of AuNPs via clathrin-mediated endocytosis in GBM cells. Moreover, after AuNP-enabled photothermal therapy, which is dependent on the AuNPs’ ability to convert near-infrared light, increased cell death (by up to 5-fold) and apoptosis of NHE9-overexpressing GBM cells, in comparison to control cells, were observed. As neither laser therapy or AuNPs alone did not considerably affect the viability of the tested cells, it was concluded that greater uptake of AuNPs increases the temperatures during treatment, and thus enhances cell death [32]. 

An alternative method for increasing the success of delivering drugs to brain tumor cells by overcoming the BBTB is convection-enhanced delivery (CED) [56]. This technique allows the distribution of larger molecules, otherwise unable to transit across the BBTB, through the application of a positive pressure. This promotes the dispersal of drugs and other therapeutic substances at the tumor site. Nevertheless, several drawbacks to the method have been investigated, including enhanced interstitial fluid flow within the surrounding brain tissue [57,58,59]. In gliomas, intravenous administration of AuNPs has also been shown to be superior to intratumoral infusion by CED, as the particles were found to access the tumor site better and present a greater therapeutic effect [60]. However, CED of nanoparticles remains a subject of active investigation. Notably, several studies indicate that the coating of nanoparticles with PEG (polyethylene glycol) at a higher density (two-fold or greater) than is typically used for intravenous delivery substantially improves their penetration in brain tissue when infused by CED [61,62]. Thus, it may be necessary to purpose-build nanoparticles that are specialized for optimal diffusion following CED. 

#### 2.1.2. Application of AuNPs in Radiotherapy

Apart from their chemotherapeutic deliverance abilities, AuNPs have been reported to sensitize breast, colon, cervical and also GBM cancer cells to radiotherapy [63,64,65,66]. Specifically, Au-based NPs (42 nm) irradiated with a 6 megavoltage photon beam have been found to increase the radiosensitivity of U-87 MG cells (model human GBM-derived cell line) and result in 1.5-fold higher sensitization of the tested cells [66]. Kunoh et al. (2019) found that other GBM-derived cell lines, U-251 GBM and U-251-P1 cells (similar to cancer stem cells), treated with the novel DNA-AuNPs (DNA extracted from calf thymus) were considerably more radiosensitive, which resulted in their decreased survival, further amplified by the induction of mitotic catastrophe [67]. 

Importantly, newly devised AuNPs may also serve a dual role as radiosensitizers and drug delivery vehicles. Such a promising construct is the Au-DOX@PO-ANG bioconjugate reported by He et al. (2021). Polymerosomes, modified by coupling with angiopeptide-2 (Ang-2) and subsequently loaded with AuNPs and conjugated with doxorubicin (DOX) (Au-DOX@PO-ANG) were found to present facilitated passage across the BBB and target the malignant tissue. Precise delivery of the therapeutic platform (facilitated by the presence of Ang-2) to the tumor enabled better sensitization of the cells to radiotherapy and exertion of a considerable cytotoxic effect (40% drop in cell viability after radiotherapy). Furthermore, near-infrared (NIR) imaging showed that rats co-treated with Au-DOX@PO-ANG and radiation had significantly restricted tumor growth rate, and subsequently, extended survival in comparison to the control group. Advantages of this system include high stability and lack of toxic effect on key organs, including the heart, liver, spleen, lungs or kidneys. An additional benefit of the polymersomes is their response to the given microenvironment, as they are pH sensitive. This feature further allows modulation (e.g., increase) in the concentrations of the administered drugs [65,68]. 

Besides chemotherapeutics, AuNPs may also deliver other compounds that display antitumor effects, e.g., gallic acid (GA; an antioxidant), to GBM tumors and improve the efficiency of radiotherapy. Jing et al. (2021) investigated the effect of AuNPs combined with products of GA. The results showed that U-251 GBM cells treated with GA-AuNPs (at concentrations of 100, 150 and 200 μg/mL) had decreased survival (by 29.75, 30.0 and 31.25%, respectively) relative to the control, especially on the third day of treatment, and increased apoptosis. Cell cycle arrest was observed at the S and G2/M phases. Furthermore, GA-AuNPs (at 150 and 200 μg/mL concentrations) were determined to sensitize the cells to various doses of radiation (from 2 to 12 Gray; Gy). Thus, it was suggested that GA-AuNPs administered simultaneously with radiotherapy could be effectively applied in the treatment of GBM [69].

In order to increase the uptake of AuNPs by tumor cells, reduce their toxicity and improve organ clearance, the implementation of AuNPs with a smaller core, i.e., ultra-small AuNPs (diameter < 3 nm), as radiosensitizers has also been examined [70]. Enferadi et al. (2018) demonstrated that PEG-coated ultra-small AuNPs conjugated with Cyclo (-RGDfK) (a potent, selective αvβ3 integrin inhibitor) lead to a sensitizer enhancement ratio of 1.21–1.66 and a 1.14–1.33 dose enhancement factor (DEF) in murine ALTS1C1 glioma cells [71]. Kefayat et al. (2019) explored the radiosensitizing abilities of folic acid (FA-AuNCs)- and BSA-conjugated small gold nanoclusters on C6 rat glioma cells. The FA-AuNCs effectively targeted the glioma cells and presented a DEF of 1.6, which is higher than that reported by Enferadi et al. (2018) and suggests that these nanoconstructs were internalized more effectively. Application of FA-AuNCs and simultaneous radiation therapy resulted in an almost 2-fold prolongation of survival of the tested glioma-bearing rats (the average survival time increased from 12.8 to 25 days). Importantly, the evaluation of the FA-AuNC’s toxicity did not reveal any damage to vital organs [72]. In a different approach to GBM therapy, Dong et al. (2021) considered the application of integrated pharmaceutics (D-iGSNPs) built of the BBB penetration iRGD-modified peptide, gold particles and the cell cycle regulator α-difluoromethylornithine. They observed that D-iGSNPs presented a significantly enhanced ability to penetrate the BBB and displayed improved targeting of tumor cells. In vivo experiments also showed that D-iGSNPs affected the cell cycle by promoting cells from the “S” to the “G” phase and sensitized cells to radiotherapy [73]. 

Finally, gold nanopeanuts (AuNPes), which are thermally stable up to 110 °C and present favorable drug-immobilizing properties (e.g., cisplatin), may be valuable in concurrent chemo- and radiotherapy. The characteristic shape of the nanoconstructs, which resembles a “peanut”, increases the surface area for the immobilization of chemotherapeutics enabling the delivery of a larger cargo of drugs [74].

#### 2.1.3. Photosensitizing Properties of AuNPs

Due to the numerous obstacles to successful clinical application of photodynamic therapy, which include short penetration of light or proper tissue oxygenation, innovative improvements are being sought out [75]. Alongside radiosensitizing properties, AuNPs have also been reported to present photosensitizing abilities. Battogtokh et al. (2019) investigated a glycol chitosan (GC)-coated gold nanocage (AuNC), which contained SiNC (silicon 2,3-naphthalocyanine bis, a near-infrared photosensitizer) and a cleavable peptide (cysteine-glycine-phenylalanine-leucine-glycine) or stable cysteine linkage. The 120–160 nm nanoconstructs with the cleavable peptide linkage (GC-pep@SiNC-AuNC) exhibited strong phototoxicity and inhibited tumor growth. Irradiation of cells pre-treated with GC-pep@SiNC-AuNC at 785 nm resulted in a drop in viability of U-87 MG cells, with the most significant decrease (~25%, relative to cells treated with free SiNC) observed at the dose of 5 μg/mL. Measurements of bioluminescence intensity (BI), which correlates with tumor growth, further confirmed that the GC-pep@SiNC-AuNC complex effectively inhibits tumor progression in vivo (BI of 57.6 in comparison to a BI of 340.8 in the control mice) [76]. 

Similarly, gold nanowreaths (AuNWs) with small magnetic iron oxide NPs (IONPs) attached on their surface are characterized by enhanced photothermal properties [40]. AuNWs present a diverse structure with numerous branches, connections and holes, which enables assembly with magnetic IONPs. The function of the construct was reported to be dependent on the level of glutathione (GSH) and in the presence of high GSH levels, the small magnetic IONPs detached from the AuNWs and this further improved MRI imaging. Moreover, the complex enabled more efficient photo-ablation of the targeted tumors in vivo.

In a different approach, He et al. (2021) investigated coating of AuNPs as a method of improving phototherapy. Application of the developed polymeric AuNPs that were conjugated with biotin increased cellular uptake (by 40%) and improved photothermal therapy of C6 glioma-derived cells [77]. AuNPs coated with keratin and tested in an in vitro assay (using U-87 MG cells) also exhibited exceptional biocompatibility and photothermal heating abilities [78]. Moreover, fluorescent gold nanostars (plasmonics-active AuNPs) have been considered for photothermal therapy, as they present superior photothermal properties and activate apoptosis, leading to GBM cell death [79]. Addition of a fluorescent dye (e.g., Atto 655) enables the monitoring of the effectiveness of the administered therapy [80]. Additionally, administration of plasmonic gold nanostars, combined with laser-induced phototherapy and immune checkpoint inhibitors (referred to as SYMPHONY), has been reported to extend the survival and enhance immunologic memory in CT-2A glioma-bearing mice, and thus, serve as a promising photothermal and immunotherapeutic approach [81].

#### 2.1.4. Cold Atmospheric Plasma and AuNPs

Interest in the application of cold atmospheric plasma (CAP) in oncological treatment has increased over the past decade. CAP consists of various particles, including electrons, neutrons or reactive species [82]. This ionized gas has previously been implemented in certain medical procedures (e.g., blood coagulation); however, its promising anticancer effects, including the promotion of cell detachment and apoptosis or inhibition of migration, make it a potential therapeutic tool [83]. Thus, He et al. (2018) investigated CAP treatment as a method of improvement of AuNP uptake by GBM cells and increasing their effectiveness in U-373 MG GBM cells. CAP was found to induce ATP-dependent endocytosis of AuNPs and increase death of U-373 cells by ~25-fold (in comparison to free-AuNPs). Moreover, this approach may be used as a potential drug delivery or diagnostic method, as the AuNPs accumulate in acidic intracellular vesicles, including lysosomes [84].

#### 2.1.5. Clinical Implications

The pre-clinical observations regarding gold nanoparticles are gradually being transferred to human clinical studies. A recent clinical trial involving AuNPs has been in progress (NCT03020017). In this open-label early phase I study, patients diagnosed with recurrent GBM or gliosarcoma, who are qualified for surgery, are administered the drug NU-0129, which presents brain-penetrating abilities. The therapeutic is composed of AuNP core-based nanocarriers and small interfering RNA (siRNA) spherical nucleic acids. To improve its stability, the core structure is shielded with surface-passivating oligoethylene glycol (OEG) (Figure 3, adopted from Kumthekar et al. 2021). The siRNA oligonucleotides are complementary to the Bcl2L12 gene that encodes for Bcl2-like protein 12, a member of the anti-apoptotic BCL2 family [85]. It was found that the intravenously administrated nanocomplex targets the tumor and silences the expression of the Bcl2L12 gene [86]. The advantages of NU-0129 include its ability to cross the BBB and target Bcl2L12, which in turn suppresses further growth of GBM.

### 2.2. Silver Nanoparticles (AgNPs)

Silver nanoparticles (AgNPs) are another example of metal nanoparticles that have been considered for application in GBM therapy. AgNPs may be synthesized using a variety of methods, with the most common including physical strategies (e.g., laser ablation or evaporation–condensation) or chemical reduction with the use of reducing agents [87]. They present unique catalytic activity (as reducing agents), alongside antitumor and antibacterial properties [88,89]. AgNPs have been found to also exert cytotoxic effects (Table 1). Such properties were observed in AgNPs synthesized using various polysaccharides isolated from marine algae (e.g., *Saccharina cichorioides*) when tested on C6 rat glioma cells [89]. Additionally, an anticancer and antibacterial effect was achieved using *Zizyphus mauritiana* fruit or *Kaempferia rotunda* (a medical herb) tuberous rhizome extract-mediated silver/silver chloride (Ag/AgCl) NPs. Usage of this nanotool against human GBM stem cells (GSCs) in vitro or Ehrlich ascites carcinoma cells in vivo (mice model) showed the induction of cell cycle arrest at the G2/M phase and reduction of the growth rate of GBM cells [90]. Other studies have further confirmed that AgNPs interfere with certain molecular pathways connected to inflammation or cellular repair processes, such as the mitogen-activated protein kinase (MAPK) cascade, in human glial cells [91]. 

#### 2.2.1. Application of AgNPs in Radiotherapy

Because of the limited therapeutic options and inherent radioresistance of GBMs [92], AgNPs were assessed for the potential to radiosensitize GBM cells and tumors in vitro and in vivo [93,94,95,96]. The earliest study examining radiosensitization of brain tumors by AgNPs evaluated the efficacy of intratumoral administration of AgNPs in combination with a single dose of ionizing radiation for the treatment of intracranial, C6 glioma-bearing rats [93]. AgNPs (10 or 20 μg in 10 μL of saline) were stereotactically injected directly into tumors 8 days after they were implanted. One day after AgNP injection, glioma-bearing rats received a 10 Gy radiation dose. Approximately 40% of rats treated with AgNPs (either dose) and ionizing radiation showed no evidence of tumors at 200 days post-treatment. The mean survival times were 100.5 and 98 days for the C6 glioma-bearing rats following treatment with 10 or 20 μg of AgNP plus radiation. In contrast, the mean survival times for irradiated controls, 10 and 20 μg AgNPs alone, and untreated controls were 24.5, 16.1, 19.4 and 16.4 days, respectively. 

The ability of AgNPs to radiosensitize GBM cells was compared to that of AuNPs [94]. Not only was a substantial radiation dose enhancement effect of AgNPs observed, this effect was also found to be superior to that of AuNPs. Specifically, at an equivalent 50 μM concentration and 6 Gy ionizing radiation dose, AgNPs achieved a radiation dose enhancement ratio in vitro of 1.64 compared to 1.23 for gold nanoparticles for the treatment of U-251 GBM cells. In vivo studies using mice implanted with intracranial U-251 tumors injected with AgNPs or gold nanoparticles confirmed these in vitro findings. Mice treated with AgNPs (10 μg) injected directly into tumors and 8 Gy ionizing radiation survived an average of 61.7 days compared to only 43.1 days for mice treated with AuNPs (10 μg) and ionizing radiation. Mice treated with ionizing radiation alone only survived for 35.1 days.

A subsequent study by Zhao et al. (2019) indicated that increased specificity of AgNPs for gliomas could be achieved by using 18 nm diameter, spherical AgNPs functionalized with a tumor-targeting aptamer (As1411) [95]. As1411 is a 6-base, guanine-rich oligodeoxynucleotide that binds to nucleolin expressed on cancer cell surfaces, and it exhibits anti-proliferative abilities [97,98]. The targeted AgNPs were efficiently taken up by C6 glioma cells, but not by normal mammary epithelia cells used as a non-cancer control. Additionally, the aptamer-functionalized AgNPs penetrated more deeply into three-dimensional tumor spheroids than non-targeted AgNPs. In vivo studies indicated that the accumulation of intravenously administered, aptamer-functionalized AgNPs within intracranial C6 tumors implanted in BALB/c mice was approximately 2.2 times greater than that of non-targeted AgNPs. At a non-toxic dose (1/10 the IC_50_ value), aptamer functionalized AgNPs were also more effective at radiation sensitization of C6 cells in vitro compared to non-targeted AgNPs, with a dose enhancement ratio of 1.62 for aptamer-functionalized AgNPs versus 1.31 for non-targeted AgNPs. Survival studies indicated that mice bearing intracranial C6 tumors lived for an average of 45 days when treated intravenously with aptamer-functionalized AgNPs (10 mg/kg) plus ionizing radiation (6 Gy) 6 h after AgNP injection, but only 35 days when treated with non-targeted AgNPs plus ionizing radiation, or 24 days for ionizing radiation alone. More recently, Zhao et al. (2021) also reported that gliomas may be further sensitized to radiotherapy through treatment with modified bovine serum albumin-coated AgNPs conjugated to verapamil and targeted using As1411 [99]. Verapamil is a drug used in the treatment of angina or hypertension that also has anticancer activity [98]. The AS1411- and verapamil-conjugated AgNP complexes were found to present better accumulation at the glioma tumor site and showed radiosensitizing abilities both in vitro (U-251 GBM cells) and in an in vivo model (glioma-bearing nude mice) [99]. 

Mechanistically, it remains unclear why AgNPs are superior to gold nanoparticles for radiation sensitization of GBM. It is likely that this is due to biological effects of AgNPs since the observed dose enhancement ratios are greater than what would be predicted based upon physical interactions of AgNPs with ionizing radiation [100]. The radiosensitizing effects of AgNPs may be due to increased autophagy in GBM cells following AgNP exposure [101,102]. Increased autophagy could be indicative of oxidative damage to proteins and organelles caused by AgNPs, and concerns about AgNP toxicity in general may limit their clinical potential. However, AgNPs have been shown to selectively increase oxidative protein, lipid and DNA damage in some triple-negative breast cancers at doses that do not affect normal breast epithelial cells [103,104,105,106]. Whether similar selective induction of damage is observed for the treatment of GBM with AgNPs without damaging or radiosensitizing normal brain cells remains to be determined. AgNPs were reported to be more cytotoxic to GBM cells than to normal lung fibroblasts or normal mammary epithelial cells, indicating that there is some potential for GBM specific toxicity at doses below the toxicity threshold for normal tissue [94,95]. Liu et al. (2018) also confirmed the radiosensitizing abilities of AgNPs on U-251 and C6 cells [96]. Importantly, they showed that glioma cells cultured in hypoxic conditions, which typically occur in a GBM microenvironment and stimulate the spreading of cancerous cells, are more susceptible to the radiosensitizing abilities of AgNPs. The sensitization enhancement ratios of the tested AgNPs were evaluated and reported to be 20% greater in hypoxic, than normoxic cells, respectively [96]. 

#### 2.2.2. Peptide Functionalization of AgNPs

Peptide functionalization is an especially promising strategy in tumor treatment, as it allows the targeting of specific proteins that are present or typically upregulated in cancerous cells and improves uptake of systemic AgNPs by GBM cells [107]. Lingasamy et al. (2019, 2020, 2021) investigated a construct composed of metallic AgNPs decorated with iron oxide nanoworms for targeting Tenascin-C (an ECM protein engaged in neovascularization). This AgNP-iron oxide nanoworm construct was further functionalized with novel targeting peptides (PL1 or PL3), which interact with receptor proteins upregulated in malignant tissue. Both peptides were found to be promising theranostic agents, as they effectively homed to GBM xenografts. Moreover, treatment with the Ag-based bioconjugates containing PL1 or PL3 was reported to effectively extend the survival of the orthotopic GBM-bearing nude mice [108,109,110]. 

#### 2.2.3. Cold Atmospheric Plasma and AgNPs

As mentioned above, the application of CAP has been proposed as a method for enhancing the uptake of metal-based NPs by GBM cells. When applied together with AgNPs, CAP lowered their surface charge distribution and, most importantly, promoted their internalization and accumulation in U-373 cells in vitro. Simultaneous treatment with AgNPs and CAP (exposures of 25 s and 40 s) also enhanced the cytotoxic effect of the treated GBM cells by 67-fold and 100-fold, respectively. Therefore, the combinatory usage of CAP, together with gold or silver NPs, resulted in improved antitumor effects, supporting further studies of this co-treatment method [111].

#### 2.2.4. Applications of AgNPs for Photothermal Therapy

Treatments for GBM include laser interstitial thermotherapy (LITT), a technique in which a laser is inserted directly into a GBM tumor under imaging guidance and used for thermal ablation [112]. Although effective for shrinking tumors, LITT is non-curative due to difficulties in achieving clean margins, treatment of large tumors and minimizing damage to a normal brain while treating invasive GBM cells. Depending upon their size and shape, AgNPs can be designed with plasmon resonance frequencies in the NIR spectrum, enabling them to generate heat when excited by wavelengths of light emitted by lasers currently used in interstitial thermal therapy of GBM (e.g., 808 nm and 1064 nm) [113,114,115]. One challenge to using AgNPs for photothermal therapy is that they are rapidly etched in biological environments, altering their shape and decreasing their efficiency at converting light to heat. To address this, more stable hybrid materials made of AgNPs in a TiO_2_ shell [116] or graphene-decorated AgNPs have been developed for photothermal cancer therapy [117]. The increased efficiency of light to heat conversion and more rapid deposition rate of heat enabled by these types of AgNPs can improve the delivery and localization of therapeutic doses of heat. Furthermore, heat itself is a radiosensitizer when given concurrently or immediately prior to ionizing radiation exposure [118]. Although not yet tested for use in GBM, preclinical studies using AgNPs for combined photothermal treatment and radiation sensitization of breast cancer indicate that the combined sensitizing effects of AgNPs, heat and radiation can selectively kill cancer cells without killing normal cells [115]. Furthermore, intratumoral injection of AgNPs (10 ug/cm^3^ based upon tumor volume) significantly sensitized intracranial C6 glioma tumors in rats to hyperthermia generated by the excitation of superparamagnetic iron oxide nanoparticles (3 mg/cm^3^) (see Section 2.3 below), indicating that AgNPs may also sensitize GBM to heat-based treatments [119].

#### 2.2.5. Use of AgNPs for Drug Delivery and Sensitization to Chemotherapy

As noted above, AgNPs themselves can be cytotoxic. Exposure to drug-free AgNPs decreased proliferation and caused cell death in U87 human GBM cells grown on chicken embryo chorioallantoic membranes [120] and T98G human GBM cells grown in standard plastic tissue culture bottles [91]. This cytotoxicity may be desirable for combination therapy with other agents. For example, AgNPs can sensitize U-251 GBM cells to TMZ [121]. Co-encapsulation of AgNPs and alisertib, an aurora A kinase inhibitor, in a polymeric nanoparticle resulted in a synergistic increase in the cytotoxicity of both agents for the treatment of U87 cells in vitro [122]. In the same study, conjugation of a chlorotoxin-based targeting ligand to this nanoparticle increased both its delivery and the therapeutic efficacy for the treatment of intracranial U87 tumors in mice following intravenous injection. The targeted NPs were able to significantly decrease tumor grown while neither the non-targeted nanoparticles nor any of the individual components altered tumor growth compared to untreated controls. Lastly, AgNPs coated with albendazole-loaded, menthol functionalized albumen were observed to cross the BBTB and enter intracranial C6 glioma tumors in mice [123]. Albendazole is known to inhibit glycolysis [124]. AgNPs were found to increase its toxicity in GBM models [123]. In the same study, menthol functionalization increased the passage of BSA-coated AgNPs across in vitro models of the BBB and increased tumor accumulation in vivo. Mice implanted with intracranial C6 glioma tumors lived an average of 24 days following treatment with menthol-functionalized, albendazole-loaded, BSA-coated AgNPs injected intravenously every other day. In contrast, saline-treated mice only survived for 15 days. These studies indicate that AgNPs offer both the potential to increase the delivery and therapeutic efficacy of current and experimental drugs for the treatment of GBM.

### 2.3. Non-Iron-Based Magnetic Nanoparticles (MNPs)

Magnetic NPs (MNPs) are another class of nanotools which may be useful in the treatment of tumors, including GBMs. They may induce magnetic hyperthermia after exposure to an alternating magnetic field (AMF), depending on the rotation of the MNPs (Brownian motion) and Neel relaxation [125]. Similar to AuNPs, the synthesis of MNPs can be conducted using physical methods, including the mechanical (“top-down”) approach or laser evaporation/ablation (“bottom-up” strategy). Additionally, both chemical (e.g., co-precipitation, thermal decomposition, sol-gel, microemulsion or hydrothermal methods) and biological synthesis strategies may be applied to construct MNPs [126]. There are numerous studies confirming the importance of MNPs in targeted cell labeling and tracking, MRI and photodynamic therapy [127,128,129]. MNPs may also serve as nanocarriers for targeted delivery of double-stranded RNA, antitumor peptides or chemotherapeutics [130,131,132]. Additionally, MNPs with a gold shell have recently been found to enhance the sensitivity of the novel and improved imaging method—magnetic particle imaging [133]. 

Numerous studies have considered MNPs for enhancing chemotherapy. TMZ-loaded MNPs modified using folic acid, as well as lipid-based TMZ-loaded MNPs, have also been found to exert a dual cytotoxic effect on GBM cells [134,135]. Similarly, administration of DOX-carrying magnetopolymersomes, enclosing magnetic IONPs and carboxymethylcellulose, was found to concurrently promote hyperthermia and chemotherapy [136].

In the case of MNPs, biofunctionalization or implementation of decorative molecules may further enhance their uptake by GBM cells and result in a stronger antitumor effect [137]. For example, poly-L-lysine (PLL)-coated MNPs showed increased vesicle size and enhanced uptake (by up to 12-fold) in human glioma cells in vitro [138]. Improved endocytosis of MNPs in the presence of PLL (either free or immobilized) may be attributed to the pro-internalization properties of the precursor amino acid. Similarly, coating magnetic graphene oxide NPs with polylactic-co-glycolic acid (PLGA) has been found to increase the deliverance of 5-iodo-2-deoxyuridine, a radiosensitizer, to GBM cells. This resulted in enhanced radiosensitivity of the tumor cells and led to a better therapeutic outcome of the treated rats [139]. Flower-like manganese or hyperbranched phenylboronic acid-functionalized MNPs have also been found to present high-loading drug capacity. Moreover, they may further promote intracellular heating and pH-dependent release [140,141]. 

#### 2.3.1. MNPs as Theranostic Tools

Theranostic application of nanoparticles, including AuNPs or MNPs, has been receiving much attention. A very innovative method is the application of homotypic targeting, an improved strategy based on the homotypic recognition of GBM components [142]. Such cancer cell membrane-coated NPs (CCMCNPs) composed of polymeric NPs covered with metastatic brain and breast cancer cells have recently been reported to efficiently cross the BBB [143]. This was confirmed by Tapeinos and collaborators (2019), who coated MNPs using U-251 GBM cell membranes and reported a 75% crossing efficiency in an in vitro BBB model (bEnd.3 monolayer). It was also shown (qualitative data) that the application of such modified NPs facilitates their uptake by the targeted GBM cells. Additionally, these GBM membrane-coated MNPs were found to increase intracellular temperature (by 6 °C), enhance apoptosis of the GBM cells and offer the potential to bypass the BBB [144]. Due to the numerous advantages of such biomimetic NPs, including effective drug delivery, assisted imaging of tumors or application in phototheranostics, they will most likely play an important role in future cancer studies [145].

Dufort et al. (2019) investigated treatment using gadolinium-based NPs (AGuIX; Gd_10_Si_40_C_200_N_50_O_150_H_x_) combined with TMZ for improving the efficacy of chemo- and radiotherapy. AGuIX is a NP composed of polysiloxane and gadolinium chelates, which is both safe and has been shown to effectively sensitize tumors to radiotherapy [146] (Figure 4). The reported radiosensitizing property was confirmed by Dufort et al. (2019), who showed that the administration of such nanocomplexes to glioma-bearing rats, successive treatment with TMZ for 5 days and subsequent radiotherapy results in elongated survival of the rats (by 15 days) [147]. In a parallel study by Thakare and collaborators (2019), AGuIX was functionalized with a macrocyclic chelator and NIR heptamethine cyanine dye, which allowed the monitoring of the distribution of the NP using positron emission tomography (PET), MRI and optical imaging at the same time. It was confirmed that AGuIX is eliminated by the kidneys in an in vivo model, thus preventing systemic toxicity. Furthermore, its application in multimodal imaging is especially beneficial in the context of surgical treatment, as it allows precise intraoperative localization and assessment of the GBM.

#### 2.3.2. Clinical Trials

Currently, one trial regarding the implementation of MNPs for clinical use is in progress. The study, which commenced in May 2021 (NCT04881032), aims to apply TMZ, AGuIX and radiotherapy in the treatment of newly diagnosed GBMs. The drug will be administered intravenously at three dose levels (50, 75 and 100 mg/kg), while radiation will be applied at 60 Gy in 6 weeks. The main advantages of the study will be the evaluation of the progression-free survival of patients and also the selection of the most effective dose level (Figure 5). 

### 2.4. Iron Oxide Nanoparticles (IONPs)

IONPs are nanoparticles with a wide range of applications and they may be generated in various forms, including magnetite, maghemite and hematite [148]. The MRI properties of certain IONPs, including Ferumoxide or Ferucarbotran, have been investigated in clinical studies as this group of NPs is capable of site-specific magnetic targeting [149]. They can also serve as contrast agents [150]. Additionally, in vitro and in vivo data have shown that IONPs may serve as nanocarriers for transporting chemotherapeutics (e.g., DOX) [151]. Their modification may generate potent multifunctional NPs for bypassing the BBB and delivering drugs to multi-drug-resistant (MDR) glioma cells. Magnetite NPs, which are a major subcategory of IONPs, have also been proposed for anti-GBM therapy, as these nanoparticles successfully induce hyperthermia of glioma cells in vivo [152]. 

#### 2.4.1. IONPs Loaded with Chemotherapeutics

Recently, drug-loaded IONPs stabilized with trimethoxysilylpropyl-ethylenediamine triacetic acid (EDT) were determined to release DOX at the tumor site at an accelerated rate. It was reported that the increased uptake of the chemotherapeutic by U-251 GBM cells leads to decreased proliferation and increased apoptosis and cell-death [149]. Importantly, enhanced BBB permeability of DOX was observed, especially after concurrent administration of a cadherin binding peptide and external magnetic field. Furthermore, the decreased expression of DNA repair enzymes (e.g., type II topoisomerase) and simultaneous increase in the expression of tumor suppressors, including p53, were reported. Consequently, such DOX-loaded EDT-IONPs conjugates are a promising treatment method for GBM, although it must be remembered that the success of such therapeutic strategies is unfortunately often hampered by the toxicity of nanoconstructs in patients [149]. 

#### 2.4.2. IONPs Loaded with Nucleic Acids

Apart from the transportation of drugs, IONPs also deliver therapeutic nucleic acids, such as microRNAs (miRNAs; miRs), to GBM cells. MiRs are small RNA molecules responsible for governing numerous key molecular pathways [153]. High levels of various miRs correlate with worse prognosis and reduced survival of patients [154]. Additionally, miRs participate in and promote processes related to carcinogenesis, including angiogenesis, metastasis or drug resistance, that impede the successful treatment of tumors [153,155]. Sukumar et al. (2019) investigated intranasal application of miR (miR-100 and anti-miR-21)-loaded polyfunctional gold IONPs, coated using the T7 targeting peptide, as a method of overcoming the BBB and sensitizing cells to TMZ. The survival of the orthotopic xenograft mice co-treated with the nanosystem and TMZ was significantly prolonged, while the accompanying labelling of the nanocomplex also highlights its future usefulness in bioluminescence imaging [156]. 

Another nucleic acid molecule that may be delivered using IONPs is siRNA. Wang et al. (2018) confirmed that delivery of nanoformulations comprising polyethyleneimine-coated Fe_3_O_4_ NPs and siRNA (targeting and knocking-down the expression of the repressor element 1-silencing transcription factor) results in decreased viability and migration of U87 MG and U-251 GBM cells [157]. In a different approach, Zang et al. (2020) created gene-therapy-based IONPs conjugated with siRNA (targeting glutathione peroxidase 4 and cisplatin), which were further modified with folate. The observed simultaneous increase in iron and H_2_O_2_ levels and induced programmed cell death of U87 MG and P3 GBM cells confirmed the complexes’ potential for the treatment of GBM [158]. 

#### 2.4.3. Theranostic Application of IONPs

IONPs are also considered a promising theranostic tool. Bifunctional constructs containing this nanoparticle may effectively captivate and transfer NIR light to heat, thus improving image-guided photothermal therapy of GBMs [27]. Moreover, gold-coated IONPs (Au@IONPs) have been shown to display promising thermo-radiosensitizing capabilities, as they lead to apoptosis and increase hyperthermia- and radiation-induced cytotoxicity of GBM-derived cells [159]. Wu et al. (2019) reported that combinatory treatment comprising radiotherapy and ferumoxytol (a cross-linked IONP; CLIO) coupled to azademethylcolchicine (ICT2552), a therapeutic drug [160], displays antitumor properties. The CLIO-ICT compound targeted GBM-initiating cells, which are suggested to play a role in recurrence of tumors and can also increase their resistance to radiotherapy. Additionally, the complex was found to effectively bypass the BBB and target tumor tissue, while not inducing necrosis in vital organs in mice [161]. 

Functionalized magnetic IONPs have also been found to promote targeted therapy of GBMs after simultaneous magnetic hyperthermia and photoacoustic self-guidance of the NPs [162]. The c(RGDyK) peptide PEG-ylated Fe@Fe_3_O_4_ NPs (RGD-PEG-MNPs) were prepared and subsequently tested both on U87 MG cells and in vivo. These altered MNPs were reported to decrease the viability of U87 MG cells by ~35% and increase the intensity of the photoacoustic signal in a U87 MG GBM murine model by 2.2-fold [162].

#### 2.4.4. Magnetosomes

An interesting idea is the therapeutic application of biologically synthesized magnetosomes (derived from magnetotactic bacteria), which are a subcategory of iron oxide MNPs and have excellent magnetic properties [163]. Mannucci et al. (2018) determined that intratumoral delivery of magnetosomes to U87 MG GBM-bearing mice and subsequent repeated magnetic fluid hyperthermia effectively suppressed the growth of the GBM [164]. The MNPs were found to stay at the tumor site for the duration of the study (two weeks), which increased the efficacy of recurrent exposure to AMF therapy. Moreover, the visualization of the tumor environment post-AMF and magnetic NP treatment showed that the temperature at the tumor site increased, which corresponded to the success of therapy. Nevertheless, as highlighted by the authors, different routes of administration of the NPs should be sought. Intratumoral delivery of MNPs may not be beneficial as AMF treatment often does not affect cancerous cells located at the periphery of the lesion and thus may enable the restoration of tumor growth [164].

### 2.5. Superparamagnetic Iron Oxide Nanoparticles (SPIONs)

Superparamagnetic iron oxide nanoparticles (SPIONs) are another theranostic tool that may potentially improve the diagnosis and treatment efficiency of gliomas. They display a broad array of properties, including low toxicity and simplicity of synthesis [165]. The potential important role of SPIONs in therapy of GBM results from their ability to exhibit superparamagnetism, which allows them to accumulate at the targeted cancerous tissue when subjected to a magnetic field [166]. Therefore, SPIONs can be implemented as an effective contrast agent in MRI and may also be important diagnostic tools for the assessment of the grade of the tumor (e.g., evaluation of MRI signal intensities after the administration of probes containing neuropilin-1 and ultra-small SPIONs) [167,168]. 

#### 2.5.1. SPIONs Co-Loaded with Chemotherapeutics

Due to their compatibility with MRI, SPIONs co-loaded with drugs allow the effective monitoring of tumor-targeted therapy. There has been a recent increase in interest concerning the delivery of DOX and SPIONs. It was found that polymeric NPs carrying DOX and SPIONs accumulate in the tumor tissue of the treated mice, which in effect restricts the growth rate of the glioma [169]. PEGylated SPIONs loaded with DOX and ICG were also found to inhibit the growth of the tumor, prevent weight loss of the tested C6 glioma-bearing rats and elongate their survival time, while not presenting any significant side effects. Additionally, in vitro data confirmed extended release, better cellular uptake of DOX and effective passage of the SPION-based NPs across the BBB, which allowed their accumulation at the site of the tumor [170]. Gholami et al. (2019) reported an antitumor effect of chitosan (poly-1-arginine-chitosan-triphosphate matrix) NPs loaded with DOX and SPIONs. Analysis of the drug release profile suggested a pH dependency and burst release was observed at a pH of 5.5, which may be indicative of endosomal or late endosomal pH. Uptake of the DOX/SPION-loaded chitosan NPs by C6 glioma cells indicates their important therapeutic implementation and confirms their efficiency in enhancing MRI resolution [171]. In vitro studies performed by Luque-Michel et al. (2019) also confirmed the stability, therapeutic effectiveness and MRI contrasting ability of nanoplatforms formed by encapsulation of DOX hydrochloride and SPIONs into PLGA, which were then coated using non-ionic surfactants [172]. 

An alternative approach consists of using solid lipid NPs encapsulating nutlin-3a and SPIONs. The in vitro study revealed that the nanovectors present colloidal stability and also enhance the apoptosis of cancer cells [173]. Similarly, lipid-based nanocarriers containing nutlin-3a and SPIONs, modified using Ang-2 (an oligopeptide capable of penetrating the BBB), promote the transfer of the nanocomplex across the BBB via transcytosis. Additionally, hyperthermia mediated by the application of an AMF was found to result in a proteolytic enzyme leakage that led to the activation of several apoptotic pathways and augmented the antitumor effect [174,175].

Alongside DOX and nutlin-3a, paclitaxel is another chemotherapeutic agent presently being tested in combination with SPIONs. The constructed PTX- and SPION-loaded PEGylated PLGA-based NPs were found to have similar properties to the aforementioned findings, including accumulation at the tumor site in the brain tissue of the tested mice, disruption of the BBB in proximity of the GBM, prolonged their survival and, most importantly, the constructs were found to be non-toxic [176]. 

#### 2.5.2. “NanoPaste” Technique

Interestingly, SPIONs can be applied in the “NanoPaste” technique to coat the inner cavity wall after the resection of the tumor. Grauer et al. (2019) co-treated GBM patients with recurrent tumors using the “Nanopaste” technique and intracavitary thermotherapy. This novel method has various benefits, including easier distribution of SPIONs precisely at the tumor site and limitation of the number of injection sites. Six GBM patients with recurrent tumors underwent treatment via this method, i.e., after the resection of the tumor, SPIONs were injected precisely to the tumor site using the “Nanopaste” technique and subsequently, the patients received six 1 h-long sessions of hyperthermia. It was found that the procedure was beneficial and promoted an anti-GBM reaction in the patients, including the induction of an inflammatory response. Nevertheless, four out of six patients had further operations due to the significant accumulation of the NPs in the tumor cavity. Thus, the authors highlight the need for further tests before this method may be safely applied on a larger scale [177]. 

#### 2.5.3. Ultra-Small SPIONs

Another class of SPIONs, which have very promising properties, are ultra-small superparamagnetic iron oxide NPs (USPIOs). They are characterized by a very reactive surface and have a <50 nm diameter, which facilitates their intratumoral dispersal [178]. Similar to SPIONs, their effectiveness in the evaluation and visualization of GBM has been especially highlighted, as they serve a dual role in the diagnosis and grading of gliomas. Their ability to serve as contrast agents allows the determination of the grade of tumor, as increased contrast is considered to be associated with a higher-grade tumor [179]. It was recently reported that the properties of USPIOs could be enhanced by modification with citric acid and subsequent conjugation with lactoferrin. These improved polygonal USPIOs exhibited satisfactory distribution and biocompatibility in numerous studies, including cytotoxicity, migration and blood biochemistry analyses [168]. PEGylated USPIOs may also be conjugated with Ang-2, which promotes their passage across the BBB, allowing the nanoprobe to be delivered to GBM and improve positive contrast in MRI. These findings confirm that USPIOs may not only serve as a diagnostic tool, but also aid in the resection of GBM [180].

### 2.6. Quantum Dots (QDs)

Quantum dots (QDs) are small (~2–10 nm) luminescent semiconductor nanocrystals that combine the merits of the aforementioned nanoparticles including the induction of a cytotoxic effect, effective penetration of the BBB and deliverance of drugs to the tumor site [181,182]. The spectrum of properties of QDs, their high tumor selectivity and their potential as fluorescence probes opens new possibilities for image-guided drug delivery and resection of gliomas [183,184]. Their photophysical properties may also facilitate bioimaging [185]. Additionally, as shown on the example of gold quantum dots (AuQDs), they can suppress metastasis of cancer cells and spheroid cell growth due to the inhibition of CTNNB1 (catenin beta 1) signaling, thus restricting the malignant properties of glioma stem-like cells [186,187]. Treatment with AuQDs, similar to AuNPs and AgNPs, has also been found to enhance the effectiveness of CAP [187]. Concurrent administration of AuQDs and CAP activated Fas/TRAIL cell death receptor pathways, resulting in a dual mechanism of cytotoxicity in GBM cells [187]. 

Alongside AuQDs, graphene quantum dots (GQDs) can be used for GBM therapy, as non-functionalized GQDs exhibit biocompatibility and low toxicity, while GQDs displaying light responsivity effectively target GBM cells and synergistically enhance the effectiveness of DOX [188,189,190]. GQDs placed on AuNPs may be used as sensing probes for the recognition of glioma cells due to their excellent selectivity and stability [191] (Table 1). 

#### 2.6.1. Dual Quantum Dots

Carbon nitride dots (CNQDs), nitrogen and boron dual-doped GQDs (N-B-GCDs) and functionalized fluorescent QDs also offer great potential for bioimaging studies [192,193,194]. N-B-GQDs exhibit few side effects, favorable second NIR window imaging and effective transfer of NIR light to heat, resulting in possible biomedical applications of these QDs in phototherapy [192]. Fluorescent dots were found to dissociate within lysosomes after endocytic uptake and activate cellular apoptosis, triggering a cytotoxic effect in U-87 MG GBM cells [195]. Similarly, CNDQs were observed to distribute to lysosomes 6 h after incubation and also selectively target pediatric GBM cells, without affecting normal human embryonic kidney (HEK293) cells after conjugation with transferrin protein [196,197,198]. Interestingly, CNQDs prepared from spices (red chili, black pepper, cinnamon, turmeric) displayed higher uptake and toxicity in LN-229 GBM cells than normal human kidney cells [199]. Although the exact mechanism regulating greater accumulation of the nanomolecules in the cytoplasm of GBM cells is unknown, the authors suggest that increased cytotoxicity of the compound may be the result of enhanced production of ROS.

Selective targeting using CNQDs conjugated with transferrin and gemcitabine (a chemotherapeutic) was proposed for the treatment of GBM in pediatric patients. The novelty of the proposed strategy was the delivery of gemcitabine across the BBB, as it has not been previously applied in the therapy of tumors of the CNS [197]. Triple-conjugated carbon dots, containing a targeted ligand and anticancer drugs, are also a promising alternative to dual drug delivery nanohybrids. Carbon dots, modified with transferrin, TMZ and epirubicin (a chemotherapeutic) presented a significantly higher cytotoxic effect on GBM cells than dual nano-drug systems containing the same chemotherapeutics when applied alone [200]. 

#### 2.6.2. Drug-Conjugated Quantum Dots and Multi-Drug Resistance

Discrepancies between the physiological pH and that of the tumor environment allow the application of a fluorescent pH-responsive drug delivery system based on PEGylated MoS2 QDs carrying DOX. This nanosystem possesses favorable theranostic properties, including limited toxicity, high stability and pH-dependent release of DOX [201]. Similarly, novel QD—biopolymer—drug nanocomplexes containing ZnS-QDs conjugated to DOX have been designated for bioimaging studies and the deliverance of drugs to GBM cells in vitro [202].

As underlined before, MDR of cancer cells is a major concern in all nanohybrid-based treatment strategies. Regarding the administration of functionalized QDs, the uptake of the fluorescent nanocarriers has been reported to differ depending on the drug-resistance profile of the cells [203]. Thus, the application of nanocarriers that are superior emitters for cell labeling and two-photon imaging may be considered, such as the recently described 2D cadmium chalcogenide nanoplatelets that display a 10 times greater two-photon absorption coefficient and better fluorescence response than QDs [204].

**Table 1 biomedicines-10-01598-t001:** Summary of metal-based therapeutic nanoparticles studied for GBM treatment.

Nanocarrier	Therapeutic Agent	Combinatory Treatment	Theranostics	In Vitro	In Vivo	Outcome	Refs.
**Gold NPs (AuNPs)**	AuNPs with TMZ; Ang-2-coupled polymerosomes with DOX; Glycol chitosan-coated gold nanocage with SiNC; AuNPs conjugated with biotin; AuNPs coated with keratin; Fluorescent gold nanostars; AuNP-based nanoplatforms; PEG-coated ultra-small AuNPs; AuNPs with CED	TMZ-AuNPs with an antibody; Folic acid- and BSA-conjugated gold nanoclusters; DNA-AuNPs; Integrated pharmaceutics; Plasmonic gold nanostars with phototherapy	Gold nanourchins; Gold nano-wreaths; Peptide-decorated AuNPs; Polyethylenimine-entrapped AuNPs	U-87 MG cells; C6 glioma cells; U-251; Murine gliosarcoma 9L	B6D2F1 mice; GL261 murine C57BL/6 model; Murine ALTS1C1 glioma; CT-2A glioma-bearing mice/rats	↑ immune response; bypassing of BBB; ↓ risk of drug resistance/toxicity; ↑ survival of rats and mice; ↓ GBM tumor growth; ↑ radiosensitization of GBM cells; pH sensitivity; ↑ photosensitization; ↑ apoptosis; ↑ chemo- and phothermal therapy; photoablation; ↑ targeted imaging and treatment of GBM tumors	[32,36,38,40,43,44,49,54,55,56,60,61,62,65,66,67,68,71,72,73,76,77,78,79,81]
**AuNPs**	AuNPs with gallic acid; Gold nanopeanuts	Cold atmospheric plasma with AuNPs	+	U-251; U-373 MG cells	-	↑ apoptosis; ↑ cell death post-radiotherapy; drug immobilization; ↑ cell cycle arrest at the S and G2/M phases; ↑ internalization of the nanocomplex	[69,74,84]
**Silver NPs (AgNPs)**	AS1411 and verapamil-conjugated AgNPs; PEG and aptamer AS1411-functionalized AgNPs; AgNPs coated with albendazole-loaded albumen	Cold atmospheric plasma with AgNPs; LITT and AgNPs; Alisertib and AgNPs; AgNPs and ionizing radiation	Peptide functionalization (targeting peptides PL1 and PL3)	C6 glioma cells; U-87 MG; U-373 GBM; U-251 glioma cells; T98G GBM	C6 glioma mice; U-87 mice model	↑ radiosensitization of GBM cells; ↑ apoptosis; ↑ survival of mice; ↑ internalization; ↑ cytotoxic effect ↑ cell cycle arrest at G2/M phase; ↑ efficiency of photothermal therapy (LITT); ↑ sensitization to TMZ and heat-based treatment	[91,93,94,96,97,98,99,108,109,110,111,116,117,119,120,121,123]
**Magnetic NPs**	MNPs with TMZ; Double-stranded RNA; PLGA-coated magnetic graphene oxide NPs; PLL-coated MNPs; Hyperbranched phenylboronic acid-functionalized MNPs	DOX-carrying magnetic IONPs and carboxymethylcellulose; Gadolinium-based NPs with TMZ	GBM-coated magnetic NPs	C6 GBM cells; U-251 MG; U-87 MG; U-118 MG	C6 rat glioma; murine glioma G422; 9L glioma-bearing rats	↑ hyperthermia/intracellular heating; ↑ apoptosis; ↑ cytotoxic effect; ↑ chemotherapy; ↑ targeted delivery; ↑ delivery of radiosensitizers; ↑ uptake by glioma cells; ↑ pH-dependent release; effective passage across the BBB; ↑ survival of mice	[134,135,136,138,139,140,141,144,147]
**Iron oxide NPs (IONPs)**	DOX-loaded IONP stabilized with EDT; siRNA targeting glutathione peroxidase 4 and cisplatin	-	-	U-251; U-87 MG; PC GBM	-	↓ proliferation; ↓ expression of DNA repair enzymes; ↑ apoptosis; ↑ expression of tumor suppressors; ↑ BBB permeability; ↑ programmed cell death of GBM cells	[149,158]
**Au-IONPs**	miR (miR-100 and anti-miR-21)-loaded polyfunctional Au-IONPs	-	Au-coated IONPs combined with hyperthermia and radiotherapy	U-87 MG	U-87 MG; nude mice model (Nu/Nu)	↑ survival of mice; bioluminescence imaging; ↑ apoptosis; ↑ radiation- and hyperthermia-induced cytotoxicity	[156,157,158,159]
**Magnetite, cross-linked and bifunctional IONPs**	Polyethylene-imine-coated Fe3O4 NPs and siRNA (targeting and silencing the repressor element 1-silencing transcription factor)	RT and cross-linked IONPs coupled to azademe-thylcolchicine	NPs with a magnetite core and a fluorescent carbon shell; RGD-PEG-MNPs	U-87 MG; U-251; C6 GBM cells	NSG^TM^ and C57BL/6J mice	↓ cell viability; ↓ proliferation; ↓ migration; ↓ tumor size; ↓ GBM-initiating cells; ↑ survival of mice; ↑ imaging-guided photothermal therapy	[152,157,161,162]
**Magnetosomes**	-	Magnetic fluid hyperthermia	-	U-87 MG	U-87 MG orthotopic model	↓ growth rate of tumor	[164]
**SPIONs**	Polymeric/chitosan NPs co-loaded with DOX and SPIONs; Lipid-based NPs loaded with nutlin-3a and SPIONs; PEGylated NPs loaded with SPIONs and PTX/or ICG; NanoPaste	+	PLGA-encapsulated SPIONs and DOX; PEGylated USPIOs with Ang-2	U-87 MG; 9L/LacZ; BBB model	C6 glioma-bearing rats; U-87 MG orthotopic model	↓ growth rate of tumor; ↑ survival of mice; ↓ weight loss of mice; ↑ MRI resolution; ↑ effective passage across BBB; ↑ apoptosis	[168,169,170,171,172,173,174,175,176,177,180]
**Quantum dots; Gold (AuQD) and graphene (GQD) quantum dots; PEGylated MoS2 QDs**	AuQDs; GQDs displaying light responsivity	Cold atmospheric plasma with AuQDs	Graphene quantum dots on AuNPs; PEGylated MoS2 QDs loaded with DOX; DOX-conjugated ZnS-QDs	Glioma stem-like cells; U-87	-	↓ metastasis; ↓ spheroid cell growth; ↑ effectiveness of CAP and DOX; ↑ dual cytotoxicity; sensing probes (stability and selectivity) ↑ pH-dependent release of the chemotherapeutic; bioimaging studies; ↑ delivery of drugs; high stability	[186,187,191,195,201,202]
**Carbon nitride dots (CNQDs); Nitrogen and boron dual-doped GQDs (N-B-GQDs)**	CNQDs conjugated with transferrin and gemcitabine	Transferrin-, TMZ- and epirubicin-modified carbon dots	N-B-GQDs	U-87 MG; SJGBM2; CHLA266; CHLA200	-	↑ NIR light to heat transfer, imaging, phototherapy; ↑ apoptosis, cytotoxicity; ↑ selective targeting of GBM cells; ↑ gemcitabine delivery across the BBB	[197,200,204]

## 3. Conclusions

Although extraordinary advances in the field of cancer research have been made, there is still no cure for glioblastoma. Therefore, the major focus, which is being addressed by various research groups worldwide, is the discovery of an effective treatment strategy against this tumor. Currently, one of the most promising molecular strategies in the treatment of GBMs includes the usage of metallic NPs as targeted and efficient drug delivery systems. The most researched approaches are combinatory strategies, including bio-conjugated peptides, drugs and other various metals as decorative molecules, which enables attainment of multiple advantages against GBM cells. The present studies also focus on the efficient passage of nanocarriers across the BBB, and better targeting, penetration and distribution of therapeutic agent(s) at the tumor site. One of the promising approaches is the intranasal delivery of metal NPs. This route allows the delivery of various anticancer nano-based complexes to the CNS, including gold-iron oxide NPs. It is expected that drug-loaded nanocarriers will cross the olfactory epithelium by a paracellular or transcellular pathway and release drugs within the brain.

In conclusion, the described methods of using gold, silver, non-iron magnetic, iron oxide, superparamagnetic iron oxide NPs and quantum dots in GBM therapy intertwine and complement each other, while the array of different methods of their application confirms that the treatment of GBM is an exceptionally complex and challenging problem.

## Figures and Tables

**Figure 1 biomedicines-10-01598-f001:**
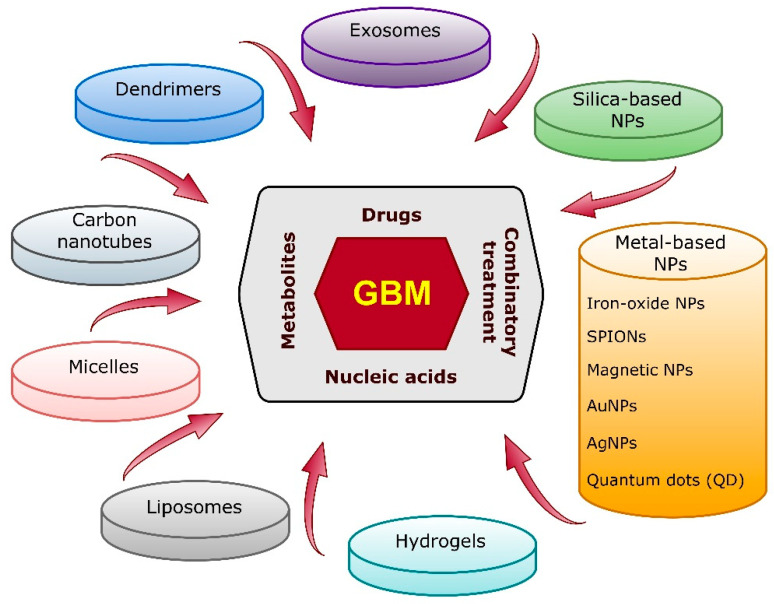
Most-studied nano-based therapeutic agent candidates for the treatment and diagnosis of GBM. NPs—nanoparticles.

**Figure 2 biomedicines-10-01598-f002:**
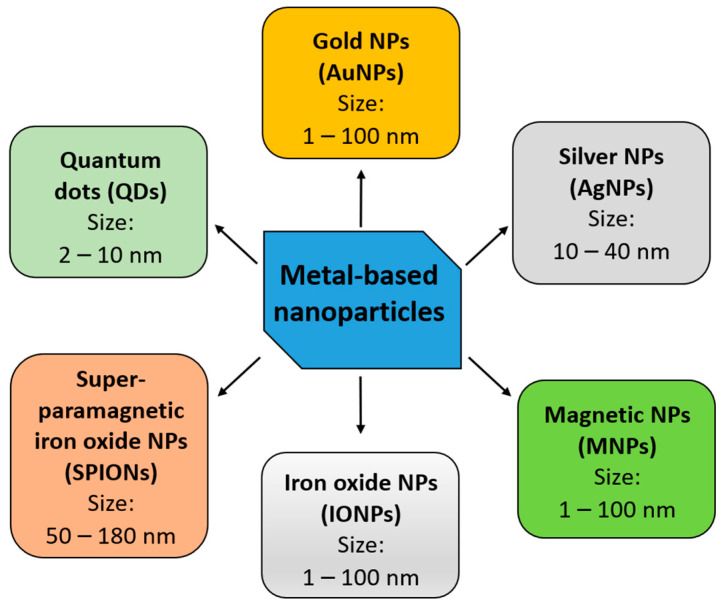
The major categories of metal-based nanoparticles currently studied for application in GBM therapy. NPs—nanoparticles.

**Figure 3 biomedicines-10-01598-f003:**
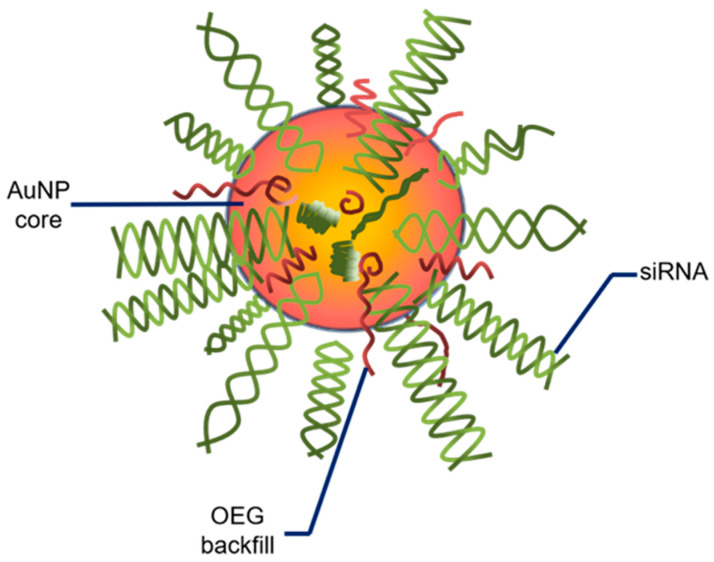
Schemata of a spherical nucleic acid. AuNP—gold nanoparticle; siRNA—small interfering RNA; OEG—oligoethylene glycol. Adopted from Kumthekar P. et al. 2021 (Sci Transl Med. 2021; 13(584):eabb3945).

**Figure 4 biomedicines-10-01598-f004:**
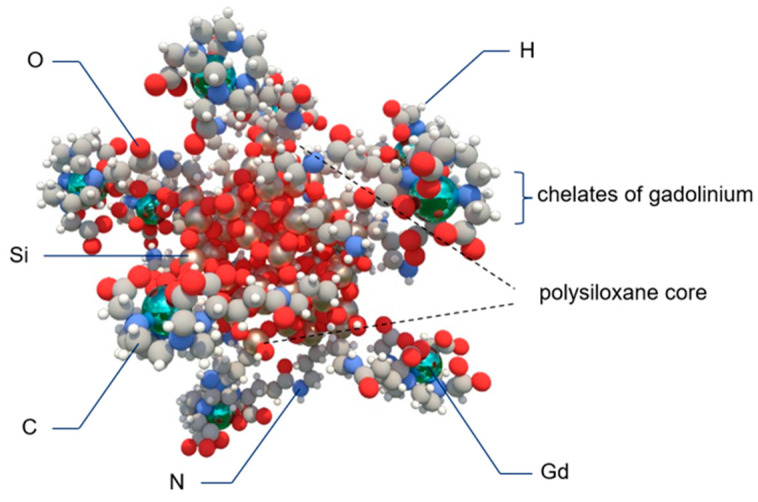
Schematic representation of AGuIX (Gd_10_Si_40_C_200_N_50_.O_150_H_x_; size: <5 nm). Polysiloxane core is built of atoms of silica (Si; grey), oxygen (O; red), hydrogen (H; white); carbon (C, grey), nitrogen (N; blue) and surrounded by covalently grafted chelates of gadolinium (Gd; green). Modified from: Verry C. et al. 2019 (BMJ Open. 2019; 9(2):e023591).

**Figure 5 biomedicines-10-01598-f005:**
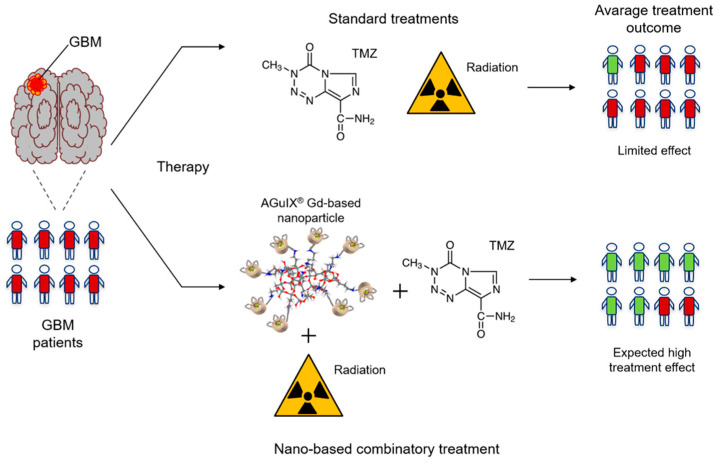
Schematic representation of the effects of AGuIX-based treatment. Patients treated simultaneously with AGuIX, TMZ and radiation are expected to do better than patients only administered with TMZ and radiation. GBM—glioblastoma; TMZ—temozolomide; AguIX—Gd-based nanoparticle.

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
