# Peer review of "Metal-Based Nanostructured Therapeutic Strategies for Glioblastoma Treatment—An Update"

_biomedicines, 2022, doi:10.3390/biomedicines10071598_

Round 1

Reviewer 1 Report

In this review paper, Gawel and the authors described the current status of metal-based nanoparticle therapy in gliblastoma in a very detailed and comprehensive manner. Overall, enthusiasm for this review is very high as a cure of glioblastoma remains challenging and identifying effective drugs for a given glioblastoma patient is highly important. As the manuscript raises an important new possibility of using new therapeutics based on metal-based nanoparticles, I would like to recommend its publication in Biomedicines. It is an exception that I feel there is little to be modified or added. 

Reviewer 2 Report

The review article entitled „Metal-based nanostructured therapeutic strategies for glioblastoma treatment – an update” focuses on the beneficial properties of commonly applied metal-based nanoparticles suitable for the treatment of glioblastoma multiforme. The review is conscious, well-written and contains numerous up-to-date information, which can be useful for the readers. I have only some minor concerns regarding to the content!

I would have liked to have read about general preparation methods of different metal-based nanoparticles; therefore, I would recommend to the authors to add this missing information!

It would be also interesting for the readers, if the authors can provide information about the possible alternative delivery routes, which can be suitable for targeting GBM with the mentioned nanoparticles bypassing the BBB (e.g., nose-to-brain delivery)!

Reviewer 3 Report

This article is focus on the interaction of metal nanoparticles with various forms of electromagnetic radiation for use in photothermal, photodynamic, magnetic hyperthermia, and ionizing radiation sensitization applications. The topic is very well described and the work is well organized in several paragraphs. The english language is fine.